# Morphological Inflection Generation
# with Hard Monotonic Attention

## Abstract

We present a neural model for morphological inflection generation which employs a hard attention mechanism, inspired by the nearly-monotonic alignment commonly found between the characters in a word and the characters in its inflection. We evaluate the model on three previously studied morphological inflection generation datasets and show that it provides state of the art results in various setups compared to previous neural and non-neural approaches. Finally we present an analysis of the continuous representations learned by both the hard and soft attention (Bahdanau et al., 2014) models for the task, shedding some light on the features such models extract.

## 1 Introduction

Morphological inflection generation involves generating a target word (e.g. "härtestem", the German word for "hardest"), given a source word (e.g. "hart", the German word for "hard") and the morpho-syntactic attributes of the target (POS=adjective, gender=masculine, type=superlative, etc.).

The task is important for many down-stream NLP tasks such as machine translation, especially for dealing with data sparsity in morphologically rich languages where a lemma can be inflected into many different word forms. Several studies have shown that translating into lemmas in the target language and then applying inflection generation as a post-processing step is beneficial for phrase-based machine translation (Minkov et al., 2007; Toutanova et al., 2008; Clifton and Sarkar, 2011; Fraser et al., 2012; Chahuneau et al., 2013)

and more recently for neural machine translation (García-Martínez et al., 2016).

The task was traditionally tackled with hand engineered finite state transducers (FST) (Koskenniemi, 1983; Kaplan and Kay, 1994) which rely on expert knowledge, or using trainable weighted finite state transducers (Mohri et al., 1997; Eisner, 2002) which combine expert knowledge with data-driven parameter tuning. Many other machine-learning based methods (Yarowsky and Wicentowski, 2000; Dreyer and Eisner, 2011; Durrett and DeNero, 2013; Hulden et al., 2014; Ahlberg et al., 2015; Nicolai et al., 2015) were proposed for the task, although with specific assumptions about the set of possible processes that are needed to create the output sequence.

More recently, the task was tackled using neural sequence to sequence learning over character sequences with impressive results (Faruqui et al., 2016). The vanilla encoder-decoder models as used by Faruqui et al. compress the input sequence to a single, fixed-sized continuous representation. Instead, soft-attention based sequence to sequence learning paradigm (Bahdanau et al., 2014) allows directly conditioning on the entire input sequence, and was utilized for morphological inflection generation with great success (Kann and Schütze, 2016b,a).

However, the sequence-to-sequence models require large training sets in order to perform well: their performance on the relatively small CELEX dataset is inferior to the latent variable WFST model of Dreyer et al. (2008). Interestingly, the neural WFST model by Rastogi et al. (2016) also suffered from the same issue on the CELEX dataset, and surpassed the latent variable model only when given twice as much data to train on.

We propose a model which handles the above issues by directly modeling an almost monotonic alignment between the input and output character

sequences, which is commonly found in the morphological inflection generation task (e.g. in languages with suffixing morphology). The model consists of an encoder-decoder neural network with a dedicated control mechanism: in each step, the model attends to a single input state and either writes a symbol to the output sequence or advances the attention pointer to the next input state from the bi-directionally encoded sequence, as described visually in Figure 1.

This modeling suits the natural monotonic alignment between the input and output, as the network learns to attend to the relevant inputs before writing the output which they are aligned to. The encoder is a bi-directional RNN, where each character in the input word is represented using a concatenation of a forward RNN and a backward RNN states over the word's characters. The combination of the bi-directional encoder and the controllable hard attention mechanism enables to condition the output on the entire input sequence. Moreover, since each character representation is aware of the neighboring characters, non-monotone relations are also captured, which is important in cases where segments in the output word are a result of long range dependencies in the input word. The recurrent nature of the decoder, together with a dedicated feedback connection that passes the last prediction to the next decoder step explicitly, enables the model to also condition the current output on all the previous outputs at each prediction step. The hard attention mechanism allows the network to jointly align and transduce while using a focused representation at each step, rather then the weighted sum of representations used in the soft attention model. In contrast to previous sequence-to-sequence work, we do not require the training procedure to also learn the alignment. Instead, we use a simple training procedure which relies on independently learned character-level alignments, from which we derive gold transduction+control sequences. The network can then be trained using straightforward cross-entropy loss.

To evaluate our model, we perform extensive experiments on three previously studied morphological inflection generation datasets: the CELEX dataset (Baayen et al., 1993), the Wiktionary dataset (Durrett and DeNero, 2013) and the SIG-MORPHON2016 dataset (Cotterell et al., 2016). We show that while our model is on par with

or better than the previous neural and non-neural state-of-the-art approaches, it also performs significantly better with very small training sets, being the first neural model to surpass the performance of the weighted FST model with latent variables specifically tailored for the task by Dreyer et al. (2008). Finally, we analyze and compare our model and the soft attention model, showing how they function very similarly with respect to the alignments and representations they learn, in spite of our model being much simpler. This analysis also sheds light on the representations such models learn for the morphological inflection generation task, showing how they encode specific features like a symbol's type and the symbol's location in a sequence.

To summarize, our contributions in this paper are three-fold:

1. We present a hard attention model for nearly-monotonic sequence to sequence learning, as common in the morphological inflection setting.

2. We evaluate the model on the task of morphological inflection generation, establishing a new state of the art on three previously-studied datasets for the task.

3. We perform an analysis and comparison of our model and the soft-attention model, shedding light on the features such models extract for the inflection generation task.

## 2 The Hard Attention Model

We would like to transduce an input sequence, $x_{1:n} \in \Sigma_x^*$ into an output sequence, $y_{1:m} \in \Sigma_y^*$, where $\Sigma_x$ and $\Sigma_y$ are the input and output vocabularies, respectively. Imagine a machine with read-only random access to the encoding of the input sequence, and a single pointer that determines the current read location. We can then model sequence transduction as a series of pointer movement and write operations. If we assume the alignment is monotone, the machine can be simplified: the memory can be read in sequential order, where the pointer movement is controlled by a single "move forward" operation (step) which we add to the output vocabulary. We implement this behavior using an encoder-decoder neural network, with a control mechanism which determines in each step of the decoder whether to write an

output symbol or promote the attention pointer the next element of the encoded input.

## 2.1 Model Definition

In prediction time, we seek the output sequence $y_{1:m} \in \Sigma_y^*$, for which:

$$y_{1:m} = \arg\max_{y' \in \Sigma_y^*} p(y'|x_{1:n}, f) \qquad (1)$$

Where $x \in \Sigma_x^*$ is the input sequence and $f = \{f_1, ..., f_l\}$ is a set of features influencing the transduction task (in the inflection generation task these would be the desired morpho-syntactic features of the output sequence). Given a nearly-monotonic alignment between the input and the output, we replace the search for a sequence of letters with a sequence of write and step actions $s_{1:q} \in \Sigma_s^*$, where $\Sigma_s = \Sigma_y \cup \{step\}$. This sequence is a series of step and write actions required to go from $x_{1:n}$ to $y_{1:m}$ according to the monotonic alignment between them. In this case we define: [1]

$$s_{1:q} = \arg\max_{s' \in \Sigma_s^*} p(s'|x_{1:n}, f)$$
$$= \arg\max_{s' \in \Sigma_s^*} \prod_{s'_i \in s'} p(s'_i|s'_1...s'_{i-1}, x_{1:n}, f) \qquad (2)$$

we can then estimate this using a neural network:

$$s_{1:q} = \arg\max_{s' \in \Sigma_s^*} NN(x_{1:n}, f, \Theta) \qquad (3)$$

Where the network's parameters $\Theta$ are learned using a set of training examples. We will now describe the network architecture.

## 2.2 Network Architecture

**Notation** We use bold letters for vectors and matrices. We treat LSTM as a parameterized function $\mathbf{LSTM}_\theta(\mathbf{x}_1...\mathbf{x}_n)$ mapping a sequence of input vectors $\mathbf{x}_1...\mathbf{x}_n$ to a an output vector $\mathbf{h}_n$. The equations for the LSTM variant we use are detailed in the supplementary material of this paper.

**Encoder** For every element in the input sequence: $x_{1:n} = x_1...x_n$, we take the corresponding embedding: $\mathbf{e}_{x_1}...\mathbf{e}_{x_n}$, where: $\mathbf{e}_{x_i} \in \mathbb{R}^E$.

---

[1] We note that our model (Eq 2) solves a different objective than (Eq 1), as it searches for the *best derivation* and not *the best sequence*. In order to accurately solve (1) we would need to marginalize over the different derivations leading to the same sequence, which is computationally challenging. However, as we see in the experiments section, the best-derivation approximation is effective in practice.

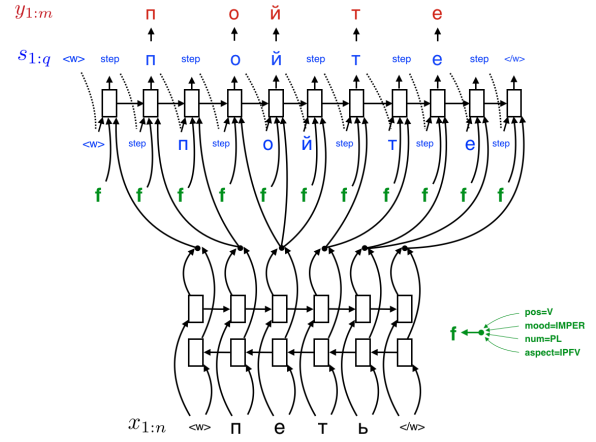

Figure 1: The hard attention network architecture. A round tip expresses concatenation of the inputs it receives. The attention is promoted to the next input element once a step action is predicted.

These embeddings are parameters of the model which will be learned during training. We then feed the embeddings into a bi-directional LSTM encoder (Graves and Schmidhuber, 2005) which results in a sequence of vectors: $\mathbf{x}_{1:n} = \mathbf{x}_1...\mathbf{x}_n$, where each vector $\mathbf{x}_i \in \mathbb{R}^{2H}$ is a concatenation of: $\mathbf{LSTM}_{\mathbf{forward}}(\mathbf{e}_{x_1}, \mathbf{e}_{x_2}, ...\mathbf{e}_{x_i})$ and $\mathbf{LSTM}_{\mathbf{backward}}(\mathbf{e}_{x_n}, \mathbf{e}_{x_{n-1}}...\mathbf{e}_{x_i})$, the forward LSTM and the backward LSTM outputs when fed with $\mathbf{e}_{x_i}$.

**Decoder** Once the input sequence is encoded, we feed the decoder RNN, $\mathbf{LSTM}_{\mathbf{dec}}$, with three inputs at each step:

1. The current attended input, $\mathbf{x}_a \in \mathbb{R}^{2H}$, initialized with the first element of the encoded sequence, $\mathbf{x}_1$.

2. A set of feature embeddings that influence the generation process, concatenated to a single vector: $\mathbf{f} = [\mathbf{f}_1...\mathbf{f}_l] \in \mathbb{R}^{F \cdot l}$.

3. $\mathbf{s}_{i-1} \in \mathbb{R}^E$, which is an embedding for the predicted output symbol in the previous decoder step.

Those three inputs are concatenated into a single vector $\mathbf{z}_i = [\mathbf{x}_a, \mathbf{f}, \mathbf{s}_{i-1}] \in \mathbb{R}^{2H+F \cdot l+E}$, which is fed into the decoder, providing the decoder output vector: $\mathbf{LSTM}_{\mathbf{dec}}(\mathbf{z}_1...\mathbf{z}_i) \in \mathbb{R}^H$. Finally, to model the distribution over the possible actions, we project the decoder output to a vector of $|\Sigma_s|$ elements, followed by a softmax layer:

$$p(s_i = c)$$
$$= softmax_c(\mathbf{W} \cdot \mathbf{LSTM}_{\mathbf{dec}}(\mathbf{z}_1...\mathbf{z}_i) + \mathbf{b}) \qquad (4)$$

**Control Mechanism** When the most probable action is $step$, the attention is promoted so $\mathbf{x}_a$ contains the next encoded input representation to be used in the next step of the decoder. The process is demonstrated visually in Figure 1.

### 2.3 Training the Model

For every example: $(x_{1:n}, y_{1:m}, f)$ in the training data, we should produce a sequence of step and write actions $s_{1:q}$ to be predicted by the decoder. The sequence is dependent on the alignment between the input and the output: ideally, the network will attend to all the input characters aligned to an output character before writing it. While recent work in sequence transduction advocate jointly training the alignment and the decoding mechanisms (Bahdanau et al., 2014; Yu et al., 2016), we instead show that in our case it is worthwhile to decouple these stages and learn a hard alignment beforehand, using it to guide the training of the encoder-decoder network and enabling the use of correct alignments for the attention mechanism from the beginning of the network training phase. For this purpose, we first run a character level alignment process on the training data. We use the character alignment model of Sudoh et al. (2013) which is based on a Chinese Restaurant Process which weights single alignments (character-to-character) in proportion to how many times such an alignment has been seen elsewhere out of all possible alignments. Once we have the character level alignment per input-output sequence pair in the training set, we deterministically infer the sequence of actions $s_{1:q}$ from it by requiring to read all the input elements aligned to an output element (using the step action) before writing it. We then train the network to predict this sequence of actions by using a conventional cross entropy loss function per example:

$$\mathcal{L}(x_{1:n}, y_{1:m}, f, \Theta) = -\sum_{s_j \in s_{1:q}} \log softmax_{s_j}(\mathbf{d}),$$

$$\mathbf{d} = \mathbf{W} \cdot \mathbf{LSTM_{dec}}(\mathbf{z}_1...\mathbf{z}_i) + \mathbf{b}$$

(5)

An alternative view of our encoder-decoder architecture is that of a *transition system* with ADVANCE and WRITE(CH) actions, where the oracle is derived from a given character-level alignment, the input is encoded using a biRNN, and the next action is determined by an RNN over the previous state/action combinations.

## 3 Experiments

We perform extensive experiments with three previously studied morphological inflection generation datasets to evaluate our hard attention model in various settings. In all experiments we report the results of the best performing neural and non-neural baselines which were previously published on those datasets. The implementation details for our models are described in the supplementary material section of this paper. The source code for our models is available on github.[2]

**CELEX** Our first evaluation is on a very small dataset, to see if our model indeed avoids the tendency to overfit with small training sets. We report exact match accuracy on the German inflection generation dataset compiled by Dreyer et al. (2008) from the CELEX database (Baayen et al., 1993). The dataset includes only 500 training examples for each of the four inflection types: 13SIA→13SKE, 2PIE→13PKE, 2PKE→z, and rP→pA which we refer to as 13SIA, 2PIE, 2PKE and rP, respectively.[3] We compare our model to three competitive models that reported results on this dataset: the Morphological Encoder-Decoder (MED) of Kann and Schütze (2016a) which is based on the soft-attention model of Bahdanau et al. (2014), the neural-weighted FST of Rastogi et al. (2016) which uses stacked bi-directional LSTM's to weigh its arcs (NWFST), and the model of Dreyer et al. (2008) which uses a weighted FST with latent-variables structured particularly for morphological string transduction tasks (LAT). Following previous reports on this dataset, we use the same data splits as Dreyer et al. (2008), dividing the data for each inflection type into five folds, each consisting of 500 training, 1000 development and 1000 test examples. We train a separate model for each fold and report exact match accuracy, averaged over the five folds.

**Wiktionary** To neutralize the negative effect of very small training sets on the performance of the different learning approaches, we also evaluate our model on the dataset created by Durrett and DeN-

---

[2] the link is removed in this version to keep the submission anonymized

[3] The acronyms stand for: 13SIA=1st/3rd person, singular, indefinite, past;13SKE=1st/3rd person, subjunctive, present; 2PIE=2nd person, plural, indefinite, present;13PKE=1st/3rd person, plural, subjunctive, present; 2PKE=2nd person, plural, subjunctive, present; z=infinitive; rP=imperative, plural; pA=past participle.

|       | 13SIA | 2PIE | 2PKE | rP   | Avg.  |
|-------|-------|------|------|------|-------|
| MED   | 83.9  | 95   | 87.6 | 84   | 87.62 |
| NWFST | 86.8  | 94.8 | 87.9 | 81.1 | 87.65 |
| LAT   | **87.5** | 93.4 | 87.4 | 84.9 | 88.3 |
| Hard  | 85.8  | **95.1** | **89.5** | **87.2** | **89.44** |

Table 1: Results on the CELEX dataset

|        | DE-N  | DE-V  | ES-V  | FI-NA | FI-V  | FR-V  | NL-V  | Avg.  |
|--------|-------|-------|-------|-------|-------|-------|-------|-------|
| DDN13  | 88.31 | 94.76 | 99.61 | 92.14 | 97.23 | 98.80 | 90.50 | 94.47 |
| NCK15  | 88.6  | 97.50 | 99.80 | 93.00 | **98.10** | **99.20** | 96.10 | 96.04 |
| FTND16 | 88.12 | **97.72** | **99.81** | 95.44 | 97.81 | 98.82 | 96.71 | 96.34 |
| YBB16  | 87.5  | 92.11 | 99.52 | 95.48 | **98.10** | 98.65 | 95.90 | 95.32 |
| Hard   | **88.87** | 97.35 | 99.79 | **95.75** | 98.07 | 99.04 | **97.03** | **96.55** |

Table 2: Results on the Wiktionary datasets

ero (2013), which contains up to 360k training examples per language. It was built by extracting Finnish, German and Spanish inflection tables from Wiktionary, used in order to evaluate their system based on string alignments and a semi-CRF sequence classifier with linguistically inspired features. We also used the expansion made by Nicolai et al. (2015) to include French and Dutch inflections as well. Their system also performs an align-and-transduce approach, extracting rules from the aligned training set and applying them in inference time with a proprietary character sequence classifier. In addition to those systems we also compare to the results of the recent neural approaches of Faruqui et al. (2016), which did not use an attention mechanism, and Yu et al. (2016), which coupled the alignment and transduction tasks.

**SIGMORPHON** As different languages show different morphological phenomena, we also experiment with how our model copes with this various phenomena using the morphological inflection dataset from the SIGMORPHON2016 shared task (Cotterell et al., 2016). Here the training data consists of ten languages, with five morphological system types (detailed in Table 3): Russian (RU), German (DE), Spanish (ES), Georgian (GE), Finnish (FI), Turkish (TU), Arabic (AR), Navajo (NA), Hungarian (HU) and Maltese (MA) with roughly 12,800 training and 1600 development examples per language. We compare our model to two soft attention baselines on this dataset: MED (Kann and Schütze, 2016b), which was the best participating system in the shared

task, and our implementation of the global (soft) attention model presented by Luong et al. (2015).

## 4 Results

On the low resource setting (CELEX), our model significantly outperforms both the recent neural models of Kann and Schütze (2016a) and Rastogi et al. (2016) and the morphologically aware latent variable model of Dreyer et al. (2008), as detailed in Table 1. It is also, to our knowledge, the first model that surpassed in overall accuracy the latent variable model on this dataset. We attribute our advantage over the soft attention model to the ability of the hard attention control mechanism to harness the monotonic alignments found in the data, while also conditioning on the entire output history which wasn't available in the FST models. Figure 2 plots the train-set and dev-set accuracies of the soft and hard attention models as a function of the training epoch. While both models perform similarly on the train-set (with the soft attention model fitting it slightly faster), the hard attention model performs significantly better on the dev-set. This shows the soft attention model's tendency to overfit on the small dataset, as it is not enforcing the monotonic assumption of the hard attention model.

On the large training set experiments (Wiktionary), our model is the best performing model on German verbs, Finnish nouns/adjectives and Dutch verbs, resulting in the highest reported average accuracy across all the inflection types when compared to the four previous neural and non-neural state of the art baselines, as detailed in Ta-

ble 2. This shows the robustness of our model also with large amounts of training examples, and the advantage the hard attention mechanism provides over the encoder-decoder approach of Faruqui et al. (2016) which does not employ an attention mechanism. Our model is also significantly more accurate than the model of Yu et al. (2016), which shows the advantage of using independently learned alignments to guide the network's attention from the beginning of the training process. As can be seen in Table 3, on the SIGMORPHON 2016 dataset our model performs better than both soft-attention baselines for the suffixing+stem-change languages (Russian, German and Spanish) and is slightly less accurate than our implementation of the soft attention model on the rest of the languages, which is now the best performing model on this dataset to our knowledge. We explain this by looking at the languages from a linguistic typology point of view, as detailed in Cotterell et al. (2016). Since Russian, German and Spanish employ a suffixing morphol-

ogy with internal stem changes, they are more suitable for monotonic alignment as the transformations they need to model are the addition of suffixes and changing characters in the stem. The rest of the languages in the dataset employ more context sensitive morphological phenomena like vowel harmony and consonant harmony, which require to model long range dependencies in the input sequence which better suits the soft attention mechanism. While our implementation of the soft attention model and MED are very similar model-wise, we hypothesize that our soft attention model results are better due to the fact that we trained the model for 100 epochs and picked the best performing model on the development set, while the MED system was trained for a fixed amount of 20 epochs (although trained on more data – both train and development sets).

## 5 Analysis

**The Learned Alignments** In order to see if the alignments predicted by our model fit the mono-

|       | suffixing+stem changes | | | circ. | suffixing+agg.+v.h. | | | c.h. | templatic | | |
|-------|-------|-------|-------|-------|-------|-------|-------|-------|-------|-------|-------|
|       | RU    | DE    | ES    | GE    | FI    | TU    | HU    | NA    | AR    | MA    | Avg.  |
| MED   | 91.46 | 95.8  | 98.84 | 98.5  | 95.47 | 98.93 | 96.8  | 91.48 | **99.3** | **88.99** | 95.56 |
| Soft  | 92.18 | 96.51 | 98.88 | **98.88** | **96.99** | **99.37** | **97.01** | **95.41** | **99.3** | 88.86 | **96.34** |
| Hard  | **92.21** | **96.58** | **98.92** | 98.12 | 95.91 | 97.99 | 96.25 | 93.01 | 98.77 | 88.32 | 95.61 |

Table 3: Results on the SIGMORPHON 2016 morphological inflection dataset. The text above each language lists the morphological phenomena it includes: circ.=circumfixing, agg.=agglutinative, v.h.=vowel harmony, c.h.=consonant harmony

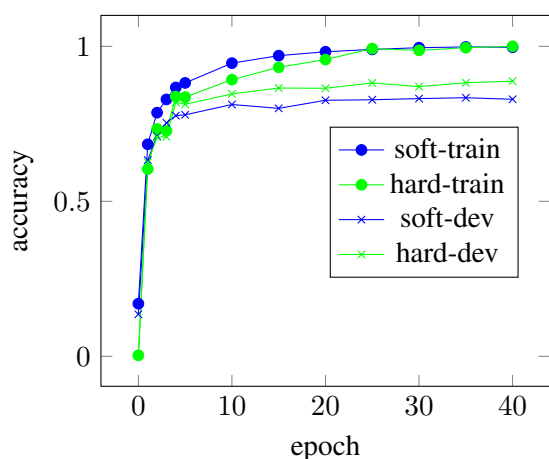

Figure 2: Learning curves for the soft and hard attention models on the first fold of the CELEX dataset

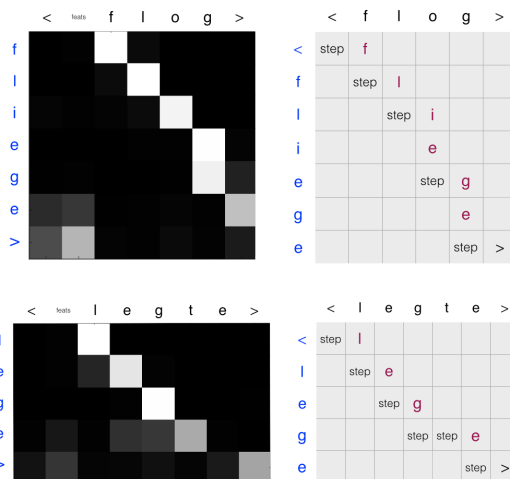

Figure 3: A comparison of the alignments as predicted by the soft attention (left) and the hard attention (right) models on examples from the CELEX dataset.

tonic alignment structure found in the data, and whether are they more suitable for the task when compared to the alignments found by the soft attention model, we examined alignment predictions of the two models on examples from the development portion of the CELEX dataset, as depicted in Figure 3. First, we notice the alignments found by the soft attention model are also monotonic, supporting our modeling approach for the task. Figure 3 (bottom-right) also shows how the hard-attention model performs deletion (legte→lege) by predicting a sequence of two step operations. Another notable morphological transformation is the one-to-many alignment, found in the top example: flog→fliege, where the model needs to transform a character in the input, *o*, to two characters in the output, *ie*. This is performed by two consecutive *write* operations after the *step* operation of the relevant character to be replaced. Notice that in this case, the soft attention model performs a different alignment by aligning the character *i* to *o* and the character *g* to the sequence *eg*, which is not the expected alignment in this case from a linguistic point of view.

**The Learned Representations** How does the soft-attention model manage to learn monotonic alignments? Perhaps the the network learns to encode the sequential position as part of its encoding of an input element? More generally, what information is encoded by the soft and hard alignment encoders? We selected 500 random encoded characters-in-context from input words in the CELEX development set, where every encoded representation is a vector in $\mathbb{R}^{200}$. Since those vectors are outputs from the bi-LSTM encoders of the models, every vector of this form carries information of the specific character with its entire context. We project these encodings into 2-D using SVD and plot them twice, each time using a different coloring scheme. We first color each point according to the character it represents (Figures 4a, 4b). In the second coloring scheme (Figures 4c, 4d), each point is colored according

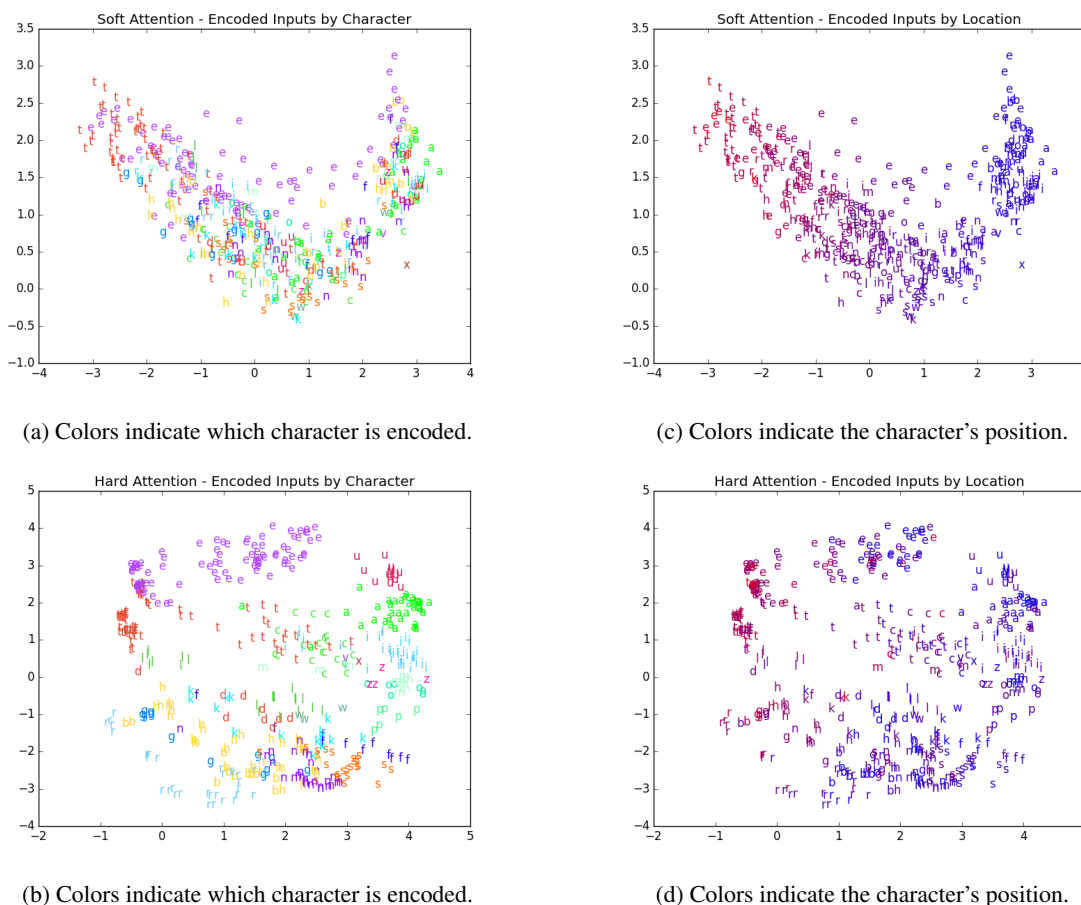

(a) Colors indicate which character is encoded.

(c) Colors indicate the character's position.

(b) Colors indicate which character is encoded.

(d) Colors indicate the character's position.

Figure 4: SVD dimension reduction to 2D of 500 character representations in context from the encoder, for both the soft attention (top) and hard attention (bottom) models.

to the character's sequential position in the word it came from, blue indicating positions near the beginning of the word, and red positions near its end.

While both models tend to cluster representations for similar characters together (Figures 4a, 4b), the hard attention model tends to have much more isolated character clusters. Figures 4c, 4d show that both models also tend to learn representations which are sensitive to the position of the character, although it seems that here the soft attention model is more sensitive to this information as its coloring forms a nearly-perfect red-to-blue transition on the X axis. This may be explained by the soft-attention mechanism encouraging the encoder to encode positional information in the input representations, which may help it to predict better attention scores, and to avoid collisions when computing the weighted sum of representations for the context vector. In contrast, our hard-attention model has other means of obtaining the position information in the decoder using the step actions, and for that reason it does not encode it as strongly in the representations of the inputs. This behavior may allow it to perform well even with fewer examples, as the location information is represented more explicitly in the model using the step actions.

## 6 Related Work

Many previous works on inflection generation used machine learning methods (Yarowsky and Wicentowski, 2000; Dreyer and Eisner, 2011; Durrett and DeNero, 2013; Hulden et al., 2014; Ahlberg et al., 2015; Nicolai et al., 2015) with assumptions about the set of possible processes needed to create the output word. Our work was mainly inspired by Faruqui et al. (2016) which trained an independent encoder-decoder neural network for every inflection type in the training data, alleviating the need for feature engineering. Kann and Schütze (2016b,a) tackled the task with a *single* soft attention model (Bahdanau et al., 2014) for all inflection types, which resulted in the best submission at the SIGMOR-PHON 2016 shared task (Cotterell et al., 2016). Another closely related work (Rastogi et al., 2016) modeled the task with a WFST in which the arcs are learned by optimizing a global loss function over all the possible paths in the state graph while modeling contextual features with bi-directional LSTM's. Another recent work (Kann et al., 2016)

proposed performing neural multi-source morphological reinflection, generating an inflection from several source forms of a word.

Previous works on neural sequence transduction include the RNN Transducer (Graves, 2012) which uses two independent RNN's over monotonically aligned sequences to predict a distribution over the possible output symbols in each step, including a null symbol to model the alignment. Yu et al. (2016) improved this by replacing the null symbol with a dedicated learned transition probability. Both models are trained using a forward-backward approach, marginalizing over all possible alignments. Our model differs from the above by learning the alignments independently, thus enabling a dependency between the encoder and decoder. While providing better results than Yu et al. (2016), this also simplifies the model training using a simple cross-entropy loss. Jaitly et al. (2015) proposed the Neural Transducer model, which is also trained on external alignments. They divide the input into blocks of a constant size and perform soft attention separately on each block. Lu et al. (2016) used a combination of an RNN encoder with a CRF layer to model the dependencies in the output sequence. A line of work on attention-based speech recognition (Chorowski et al., 2015; Bahdanau et al., 2016) proposed improvements to the attention mechanism: adding location awareness by using the previous attention weights when computing the next ones, and preventing the model from attending on too many or too few inputs using "sharpening" and "smoothing" techniques on the attention weight distributions.

## 7 Conclusion

We presented a hard attention model for morphological inflection generation. The model employs an explicit alignment which is used to train a neural network to perform transduction by decoding with a hard attention mechanism. Our model performs on par or better than previous neural and non-neural approaches on various morphological inflection generation datasets, while staying competitive with dedicated models even with very few training examples. Future work may include applying the model to other nearly-monotonic align-and-transduce tasks like abstractive summarization, transliteration or machine translation.

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

## Supplementary Material

### Training Details, Implementation and Hyper Parameters

To train our models, we used the train portion of the datasets as-is and evaluated the model which performed best on the development portion of the dataset, without conducting any specific pre-processing steps on the data. We train the models for a maximum of 100 epochs over the training set. To avoid long training time, we trained the model for 20 epochs for datasets larger than 50k examples, and for 5 epochs for datasets larger than 200k examples. The models were implemented using the python bindings of the dynet toolkit.[4] We trained the network by optimizing the expected output sequence likelihood using cross-entropy loss as mentioned in equation 5. For optimization we used ADADELTA (Zeiler, 2012) without regularization. We updated the weights after every example. We used the dynet toolkit implementation of an LSTM network with two layers, each having 100 entries in both the encoder and decoder. The character embeddings were also

vectors with 100 entries for the CELEX experiments, and with 300 entries for the SIGMOR-PHON and Wiktionary experiments. The morpho-syntactic attribute embeddings were vectors of 20 entries in all experiments. We did not use beam search while decoding for both the hard and soft attention models as it is significantly slower and did not show clear improvement in previous experiments we conducted. In all experiments, for both the hard and soft attention models, we report results using an ensemble of 5 models with different random initializations by using majority voting on the final sequences the models predicted, as reported in Kann and Schütze (2016a). This was done to perform fair comparison to the models of Kann and Schütze (2016a,b); Faruqui et al. (2016) which also performed a similar ensembling technique. For the character level alignment process we use the implementation provided by the organizers of the SIGMORPHON2016 shared task.[5]

### LSTM Equations

We used the LSTM variant implemented in the dynet toolkit, which corresponds to the following equations:

$$\mathbf{i}_t = \sigma(\mathbf{W}_{ix}\mathbf{x}_t + \mathbf{W}_{ih}\mathbf{h}_{t-1} + \mathbf{W}_{ic}\mathbf{c}_{t-1} + \mathbf{b}_i)$$
$$\mathbf{f}_t = \sigma(\mathbf{W}_{fx}\mathbf{x}_t + \mathbf{W}_{fh}\mathbf{f}_{t-1} + \mathbf{W}_{fc}\mathbf{c}_{t-1} + \mathbf{b}_f)$$
$$\widetilde{\mathbf{c}} = \tanh(\mathbf{W}_{cx}\mathbf{x}_t + \mathbf{W}_{ch}\mathbf{h}_{t-1} + \mathbf{b}_c)$$
$$\mathbf{c}_t = \mathbf{c}_{t-1} \circ \mathbf{f}_t + \widetilde{\mathbf{c}} \circ \mathbf{i}_t$$
$$\mathbf{o}_t = \sigma(\mathbf{W}_{ox}\mathbf{x}_t + \mathbf{W}_{oh}\mathbf{h}_{t-1} + \mathbf{W}_{ox}\mathbf{c}_t + \mathbf{b}_o)$$
$$\mathbf{h}_t = \tanh(\mathbf{c}_t) \circ \mathbf{o}_t$$

$$(6)$$

---

[4]https://github.com/clab/dynet

[5]https://github.com/ryancotterell/sigmorphon2016

