# Peer review of "Morphological Inflection Generation with Hard Monotonic Attention"

_ACL 2017 — decision unknown_

[Official Review · Reviewer 1 · rating 3 · confidence 4]
soundness 4 · originality 3 · clarity 5 · impact 3 · substance 4 · appropriateness 5 · meaningful comparison 2 · presentation format Poster

- Strengths:
The idea of hard monotonic attention is new and substantially different from
others.

- Weaknesses:
The experiment results on morphological inflection generation is somewhat
mixed. The proposed model is effective if the amount of training data is small
(such as CELEX). It is also effective if the alignment is mostly monotonic and
less context sensitive (such as Russian, German and Spanish).

- General Discussion:

The authors proposed a novel neural model for morphological inflection
generation which uses "hard attention", character alignments separately
obtained by using a Bayesian method for transliteration. It is substantially
different from the previous state of the art neural model for the task which
uses "soft attention", where character alignment and conversion are solved
jointly in the probabilistic model.

The idea is novel and sound. The paper is clearly written. The experiment is
comprehensive. The only concern is that the proposed method is not necessarily
the state of the art in all conditions. It is suitable for the task with mostly
monotonic alignment and with less context sensitive phenomena. The paper would
be more convincing if it describe the practical merits of the proposed method,
such as the ease of implementation and computational cost.

[Official Review · Reviewer 2 · rating 3 · confidence 3]
soundness 4 · originality 3 · clarity 5 · impact 3 · substance 4 · appropriateness 5 · meaningful comparison 2 · presentation format Oral Presentation

- Strengths: A new encoder-decoder model is proposed that explicitly takes 
into account monotonicity.

- Weaknesses: Maybe the model is just an ordinary BiRNN with alignments
de-coupled.
Only evaluated on morphology, no other monotone Seq2Seq tasks.

- General Discussion:

The authors propose a novel encoder-decoder neural network architecture with
"hard monotonic attention". They evaluate it on three morphology datasets.

This paper is a tough one. One the one hand it is well-written, mostly very
clear and also presents a novel idea, namely including monotonicity in
morphology tasks. 

The reason for including such monotonicity is pretty obvious: Unlike machine
translation, many seq2seq tasks are monotone, and therefore general
encoder-decoder models should not be used in the first place. That they still
perform reasonably well should be considered a strong argument for neural
techniques, in general. The idea of this paper is now to explicity enforce a
monotonic output character generation. They do this by decoupling alignment and
transduction and first aligning input-output sequences monotonically and
then training to generate outputs in agreement with the monotone alignments.
However, the authors are unclear on this point. I have a few questions:

1) How do your alignments look like? On the one hand, the alignments seem to
be of the kind 1-to-many (as in the running example, Fig.1), that is, 1 input
character can be aligned with zero, 1, or several output characters. However,
this seems to contrast with the description given in lines 311-312 where the
authors speak of several input characters aligned to 1 output character. That
is, do you use 1-to-many, many-to-1 or many-to-many alignments?

2) Actually, there is a quite simple approach to monotone Seq2Seq. In a first
stage, align input and output characters monotonically with a 1-to-many
constraint (one can use any monotone aligner, such as the toolkit of
Jiampojamarn and Kondrak). Then one trains a standard sequence tagger(!) to
predict exactly these 1-to-many alignments. For example, flog->fliege (your
example on l.613): First align as in "f-l-o-g / f-l-ie-ge". Now use any tagger
(could use an LSTM, if you like) to predict "f-l-ie-ge" (sequence of length 4)
from "f-l-o-g" (sequence of length 4). Such an approach may have been suggested
in multiple papers, one reference could be [*, Section 4.2] below. 
My two questions here are: 

2a) How does your approach differ from this rather simple idea?

2b) Why did you not include it as a baseline?

Further issues:

3) It's really a pitty that you only tested on morphology, because there are
many other interesting monotonic seq2seq tasks, and you could have shown your
system's superiority by evaluating on these, given that you explicitly model
monotonicity (cf. also [*]).

4) You perform "on par or better" (l.791). There seems to be a general
cognitive bias among NLP researchers to map instances where they perform worse
to
"on par" and all the rest to "better". I think this wording should be
corrected, but otherwise I'm fine with the experimental results.

5) You say little about your linguistic features: From Fig. 1, I infer that
they include POS, etc. 

5a) Where did you take these features from?

5b) Is it possible that these are responsible for your better performance in
some cases, rather than the monotonicity constraints?

Minor points:

6) Equation (3): please re-write $NN$ as $\text{NN}$ or similar

7) l.231 "Where" should be lower case

8) l.237 and many more: $x_1\ldots x_n$. As far as I know, the math community
recommends to write $x_1,\ldots,x_n$ but $x_1\cdots x_n$. That is, dots should
be on the same level as surrounding symbols.

9) Figure 1: is it really necessary to use cyrillic font? I can't even address
your example here, because I don't have your fonts.

10) l.437: should be "these"

[*] 

@InProceedings{schnober-EtAl:2016:COLING, 

  author    = {Schnober, Carsten  and  Eger, Steffen  and  Do Dinh,
Erik-L\^{a}n  and  Gurevych, Iryna},
  title     = {Still not there? Comparing Traditional Sequence-to-Sequence
Models to Encoder-Decoder Neural Networks on Monotone String Translation
Tasks},
  booktitle = {Proceedings of COLING 2016, the 26th International Conference on
Computational Linguistics: Technical Papers},
  month     = {December},
  year                                                      = {2016},
  address   = {Osaka, Japan},
  publisher = {The COLING 2016 Organizing Committee},
  pages     = {1703--1714},
  url                                               =
{http://aclweb.org/anthology/C16-1160}

}

AFTER AUTHOR RESPONSE

Thanks for the clarifications. I think your alignments got mixed up in the
response somehow (maybe a coding issue), but I think you're aligning 1-0, 0-1,
1-1, and later make many-to-many alignments from these. 
I know that you compare to  Nicolai, Cherry and Kondrak (2015) but my question
would have rather been: why not use 1-x (x in 0,1,2) alignments as in  Schnober
et al. and then train a neural tagger on these (e.g. BiLSTM). I wonder how much
your results would have differed from such a rather simple baseline. (A tagger
is a monotone model to start with and given the monotone alignments, everything
stays monotone. In contrast, you start out with a more general model and then
put hard monotonicity constraints on this ...)

NOTES FROM AC

Also quite relevant is Cohn et al. (2016),
http://www.aclweb.org/anthology/N16-1102 .

Isn't your architecture also related to methods like the Stack LSTM, which
similarly predicts a sequence of actions that modify or annotate an input?  

Do you think you lose anything by using a greedy alignment, in contrast to
Rastogi et al. (2016), which also has hard monotonic attention but sums over
all alignments?